# Matching Mechanics and Energetics of Muscle Contraction Suggests Unconventional Chemomechanical Coupling during the Actin–Myosin Interaction

**DOI:** 10.3390/ijms241512324

**Published:** 2023-08-01

**Authors:** Irene Pertici, Lorenzo Bongini, Marco Caremani, Massimo Reconditi, Marco Linari, Gabriella Piazzesi, Vincenzo Lombardi, Pasquale Bianco

**Affiliations:** PhysioLab, University of Florence, 50019 Sesto Fiorentino, Italy; irene.pertici@unifi.it (I.P.); lnzbng@gmail.com (L.B.); marco.caremani@unifi.it (M.C.); massimo.reconditi@unifi.it (M.R.); marco.linari@unifi.it (M.L.); gabriella.piazzesi@unifi.it (G.P.); pasquale.bianco@unifi.it (P.B.)

**Keywords:** skeletal muscle, chemomechanical coupling, muscle energetics, efficiency of muscle contraction, myosin–actin interaction, sarcomere-like nanomachine

## Abstract

The mechanical performances of the vertebrate skeletal muscle during isometric and isotonic contractions are interfaced with the corresponding energy consumptions to define the coupling between mechanical and biochemical steps in the myosin–actin energy transduction cycle. The analysis is extended to a simplified synthetic nanomachine in which eight HMM molecules purified from fast mammalian skeletal muscle are brought to interact with an actin filament in the presence of 2 mM ATP, to assess the emergent properties of a minimum number of motors working in ensemble without the effects of both the higher hierarchical levels of striated muscle organization and other sarcomeric, regulatory and cytoskeleton proteins. A three-state model of myosin–actin interaction is able to predict the known relationships between energetics and transient and steady-state mechanical properties of fast skeletal muscle either in vivo or in vitro only under the assumption that during shortening a myosin motor can interact with two actin sites during one ATP hydrolysis cycle. Implementation of the molecular details of the model should be achieved by exploiting kinetic and structural constraints present in the transients elicited by stepwise perturbations in length or force superimposed on the isometric contraction.

## 1. Introduction

In the sarcomere, the ~2 µm long structural unit of the cell of striated (skeletal and cardiac) muscle, force and shortening are generated by a working stroke in the myosin motors, extending in two bipolar arrays from each thick filament, during their interaction with the nearby thin, actin-containing filaments, originating at the sarcomere extremities (Figure 1).

In the myosin motor, each cycle of attachment to actin, generation of force and detachment is coupled to the hydrolysis of ATP to orthophosphate (Pi) and ADP in the catalytic site of the myosin head [1,2,3]. The working stroke in the actin-attached motor that pulls the actin filament towards the centre of the sarcomere consists in an interdomain tilting of the light chain binding domain (the lever arm) about the catalytic domain firmly attached to the actin filament, coupled with the release of the hydrolysis products: first orthophosphate (Pi) and then ADP [4,5]. In the isometric contraction, the lever arm tilting is opposed by the rising strain in the elastic elements [6], while at low load tilting of the lever arm induces filament sliding with reduced strain in the elastic elements. Detachment is promoted by the attachment of a new ATP molecule. While detached, the myosin motor hydrolyses ATP and undergoes the reversal of the working stroke [7].

## 2. Results

### 2.1. Mechanics and Energetics of Skeletal Muscle

In the half-sarcomere, the myosin motors are mechanically coupled as parallel force generators via their filament attachment (Figure 1), and the collective motor formed by the array of myosin motors and the interdigitating actin filaments is the basic functional unit of muscle. When the external load (*T*) is smaller than the isometric force generated by the motor array in each half-sarcomere (*T*_0_), the actin filaments slide past the myosin filaments and the sarcomere shortens at a velocity (*V*) that is inversely proportional to the load (force–velocity relation, *T*-*V* [8], see Figure 2A for the frog muscle). The rate of motor detachment from actin increases when, during steady shortening, motors become negatively strained, and this explains the increased rate of energy liberation (and the underlying ATPase rate) with the reduction in the load and increase in the shortening speed [1,9,10]. Faster detachment of negatively strained motors, which is attributed to a conformation-dependent acceleration of ADP dissociation followed by rapid ATP binding [11,12,13,14], prevents motors at the end of their working stroke to oppose positively strained motors, a requirement for maximization of the power and efficiency of a collective motor. The power output (*P* = *T · V*) attains its maximum value (*P*_max_) with a load *T* ~1/3*T*_0_ at which *V* is ~1/3 the maximum shortening velocity (the velocity under zero load, *V*_0_) (Figure 2B). Fitting the *T*-*V* relation with Hill’s equation (line in Figure 2A)
(*T*/*T*_0_ + *a*/*T*_0_) (*V* + *b*) = (*V*_0_ + *b*) *a*/*T*_0_,
where *a*/*T*_0_ and *b* are the distances of the horizontal and vertical asymptotes from the abscissa and the ordinate, respectively, allows to extract the information on the curvature of the relation that is larger for smaller values of the parameter *a*/*T*_0_ (an index of the radius of curvature) indicating a smaller maximum power [10].

The work made by myosin motors during the chemomechanical transduction cycle is accompanied by heat production, the rate of which increases roughly linearly with the increase in shortening velocity [8]. The rate of energy liberation *E*′ (=power + heat rate = *P* + *Q*′, Figure 2C upper line) and the rate of underlying ATP-driven chemomechanical cycles (φ, Figure 2C circles) increase with the reduction of the load [8,9,10,15,16]. *E*′ rises with the increase in *V* with a slope that progressively decreases and attains a maximum at *V* ~0.5 *V*_0_. At both *T*_0_ and *V*_0_ the power is zero and *E*′ is accounted solely by *Q*′. The efficiency of energy transduction is measured by the ratio of power over the rate of energy liberated (mechanical efficiency): ε = *P*/(*Q*′ + *P*). Alternatively, efficiency can be measured by the ratio of power over the rate of energy made available by ATP hydrolysis (thermodynamical efficiency): η = *P*/(φ·Δ*G*_ATP_), where Δ*G*_ATP_ is the free energy of the hydrolysis of one ATP, which in frog muscle at the physiological concentration of ATP and hydrolysis products is 50 kJ mol^−1^ [15]. The efficiency attains its maximum value at *V* ~0.2 *V*_0_ (Figure 2D), at which *T* is ~0.5*T*_0_ (Figure 2A).

The first line of Table 1 reports the rate of energy liberated by the *sartorius* muscle of the frog at 0 °C during production of either steady maximum isometric force (*T*_0_) or steady maximum power (*P*_max_, 22 mW g^−1^), which occurs during shortening at *V* ~1/3*V*_0_ (or T ~1/3*T*_0_). The energy rate for *P*_max_ is 45 mW g^−1^, which is ~4 times larger than at *T*_0_. Data are corrected for the energy rate not due to myosin motors. The efficiency for *P*_max_, (22/45 =) 0.49, is just below the maximum efficiency (Figure 2D).

The energetics of contraction of the frog muscle, as well as that of mammalian muscles, can be made comparable with that of demembranated fibres of mammalian muscles, in which the ATPase rate is more often measured with biochemical assays, by dividing the energy rate by the free energy of the ATP hydrolysis (Δ*G*_ATP_, 50 kJ mol^−1^) to obtain the rate of ATP hydrolysis per g (Table 1, third line). Using the concentration of myosin heads in the frog muscle (0.21 mM [17]), the ATP hydrolysed per myosin head can be calculated (fourth line), which also represents the frequency at which a myosin head undergoes the chemomechanical cycle (φ).

Notably, the rate of energy utilization during shortening at the maximum power increases to four times that of the isometric contraction, at which the power is zero and all the energy is liberated as heat. This feature underpins the efficiency of power production by contracting muscle.

A similar analysis of the mechanics and energetics of fast mammalian skeletal muscle is more complicated than for the frog muscle for a series of factors: (i) differences in myosin isoforms in different muscles of the same animal (paralog isoforms); (ii) differences in the myosin isoforms for the same muscle type in different animal models (ortholog isoforms); (iii) differences in the temperature at which data were collected, which introduce differences in the mechanokinetic parameters; (iv) heterogeneity of methodological approaches for mechanical measurements: in relatively large mammals as the rabbit, sarcomere level mechanics of the fast skeletal muscle (psoas) can only be conducted in segments of demembranated fibres; in small mammals, the whole muscle can be used for defining the mechanical performance, but sarcomere level mechanics is not accessible, and, moreover, the responses are influenced by the large end compliances; (v) heterogeneity of methodological approaches for measurements of the ATPase rate. From here on, we use the temperature of 21–25 °C as the reference value for comparing the performance of fast mammalian muscle. This temperature is lower than the in vivo temperature of mammalian muscle (~35 °C), but allows direct comparison with the parameters obtained from the half-sarcomere-like synthetic nanomachine, which is powered by heavy meromyosin fragments (HMM) of the myosin molecule (the portion of the molecule containing the motor, red in Figure 1, plus the segment of the tail emerging from the backbone of the thick filament, not shown in Figure 1 for simplicity), purified from the psoas muscle of the rabbit [18]. The paucity of measurements of the mechanical performance of skinned fibres at temperatures > 15 °C is due to the fact that at higher temperatures the sarcomeric structure, and thus the consistency of mechanical responses, cannot be maintained in subsequent contractions [19].

A careful description of the performance of fast mammalian muscle at 25 °C is reported by Ranatunga [20,21,22] using the *Extensor digitorum longus* (EDL) of the rat. The force–velocity relation of EDL from [21] is reported in Figure 3A with the force recalculated in pN per half-thick filament (htf), considering that the density of thick filaments in fast mammalian muscle is 5.7 × 10^14^ m^−2^ [23]. This unit of force is preferred to the force per cross-sectional area (defined as *T*) to make the comparison with the force produced by the synthetic nanomachine and then simulated in the model in a more straightforward manner, and is defined as *F* instead of *T* for clarity. For comparison, the *F*-*V* relation for the frog muscle fibre at 4.6 °C is reported in Figure 3C as a continuous line. Data are from Figure 1, with isometric force per half-thick filament (*F*_0_, 279 pN) calculated from the *T*_0_ = 164 kPa, considering a density of thick filaments of 5.87 × 10^14^ m^−2^ [24]. The power–force (*P-F*) relations calculated from the *F-V* relations show a maximum power (*P*_max_) of 460 aW (Figure 3B, continuous line) and 96 aW (Figure 3D, continuous line), respectively.

Table 2 reports the energetic parameters present in literature for the mammalian muscle, including the intact muscle of the mouse [26,27] and the demembranated fibres from the fast muscle of rat [28], rabbit [29,30], and human [31]. For the sake of comparison, the energy rate (*E*′) measured in intact mammalian muscle is also expressed in terms of ATP hydrolysis rate per myosin head φ, using a Δ*G*_ATP_ of 60 kJ mol^−1^ [17].

The comparison of either mechanical or energetic parameters of intact and skinned mammalian muscles makes main discrepancies emerge, which cannot be completely accounted for by considering that the temperature of the experiment is lower in the skinned fibres. For the mechanics, the curvature of the *T*-*V* relation is smaller in the intact muscle, and thus the relative maximum power is larger than in the skinned fibre: *a*/*T*_0_, a parameter that, similar to *T*_0_, has a minor dependence on temperature (Q_10_ ranging from 1 to 1.3 [10]), is generally more than two times larger in the intact muscle (0.37 [21], 0.34 [26]) than in the skinned fibre (0.07 [28], 0.14 [31], 0.09 [30]). For the energetics, the main discrepancy is in the energy rate, and thus the ATPase rate, during the isometric contraction. Assuming a Q_10_ of 2 for the underlying kinetics, *E*’_0_ and the related φ_0_, in the intact muscle cannot be made to agree with the corresponding values in the skinned fibre: φ_0_ is a factor of 10 larger in the intact mouse muscle at 20–25 °C than in the skinned rabbit fibre at 10–15 °C. The discrepancy remains large, also considering a 15–30% correction for the possible underestimate of the kinetics due to the effect of the species size [32]. Instead, a feature that is relatively well conserved between the intact and skinned preparations is the increase by a factor of ~2 of the energy rate, and ATPase rate, between isometric contraction and isotonic contraction at *V_P_*_max_: the ratio φ*_P_*_max/_φ_0_ is just below 2 in intact muscle and just above 2 in skinned fibres (Table 2). Consequently, there are also only small differences in the value of the efficiency of energy consumption for delivering the maximum power (ε*_P_*_max_ or η*_P_*_max_, ~0.3, Table 2).

### 2.2. Fitting the Performance of Muscle Contraction with a Simple Three-State Model

The mechanical performances of the fast skeletal muscle reported in Figure 3 are simulated (dashed line) with a simple three-state mechanokinetic model of the actin–myosin interaction [18,25]. As shown in Figure 4, the model assumes one detached state (D) and two different force-generating attached states (A1 and A2). The coupling between mechanical and biochemical steps and the rate functions of state transitions in the cycle have been already described in detail [18,25] and are synthetically reported in Methods (Figure 8). How the most relevant energetic features of the model are constrained by data in the literature for either frog or mammalian muscle (Table 1 and Table 2) is summarized in this paper. In isometric contraction, the rate-limiting step in the cycle is detachment from A2: under these conditions the fraction of attached motors (the duty ratio) is at a maximum, while the rate of ATP splitting per myosin head (φ_0_) is at a minimum. During steady shortening, the duty ratio decreases and the ATP splitting rate per myosin head (φ) increases, due to the increase in the rate of motor detachment following the execution of the working stroke. φ for the maximum power (φ*_P_*_max_) is higher than φ_0_ by a factor of four in frog muscle (Table 1) and two in mammalian muscle (Table 2). Under this condition, the curvature of the *F*-*V* relation (*a*/*F*_0_ ~0.36) and the resulting maximum power can be reproduced only by assuming (Model 1) that, during shortening, the attached myosin motors can rapidly regenerate the working stroke by slipping to the next actin farther from the centre of the sarcomere during the same ATPase cycle [33,34,35] (step “slip” in Figure 4) and undergoing A1′–A2′ state equilibration according to step 2 kinetics [36]. Detachment from either A1′ or A2′ (step 3′) implies ATP hydrolysis.

All the relevant mechanical and energetic parameters underlying the *F-V* relation of the half-thick filament of both the mammalian muscle (Figure 3A,B and Table 2) and the frog muscle (Figure 3C,D and Table 1) are reproduced by Model 1, as detailed in Figure 5A–C (black lines) and Table 3 (first line) for the mammalian muscle and Figure 5D–F (black lines) and Table 4 (first line) for the frog muscle. In particular, for the mammalian muscle at 25 °C, φ_0_, calculated by the flux through step 1 at *F*_0_, is 12 s^−1^ and increases to 36 s^−1^ (3·φ_0_) at *P*_max_. Similarly, for the frog muscle at 4.6 °C, φ increases by four times (from 2.3 s^−1^ to 8.6 s^−1^) from *F*_0_ to *P*_max_.

Whether the mechanical and energetic constraints summarized in Figure 3 and Table 1 and Table 2 can be fitted with a conventional mechanochemical cycle in which the ATP hydrolysis is completed within a single actin–myosin interaction is tested in two subsequent steps. In the first step (Model 2), the slipping transition (Figure 4) is removed to check its specific effects on the mechanical and energetic parameters. As shown in Table 3 and Table 4 (second line) and Figure 5 (violet), in the isometric contraction, neither the force (abscissa intercept in Figure 5A,D) nor its energetic cost (ordinate intercept in Figure 5C,F) are affected, while, regarding the shortening contraction, both *a*/*F*_0_ and *P*_max_ are reduced without a marked change in φ*_P_*_max_ (identified by the arrows in Figure 5C,F). *V*_0_ too is reduced (ordinate intercept in Figure 5A,D) so that the value of *P*_max_, with respect to the expected value (green in Figure 5B,E) is reduced to less than half (Figure 5B,E, violet and Table 3 and Table 4, second line) for the combined effect of reduction in both *a*/*F*_0_ and *V*_0_.

In the second step (Model 3), the simulation of the mechanical performance of fast skeletal muscle in the absence of slipping is pursued with ad hoc increase in the relevant rate constants for attachment and detachment in the scheme of Figure 4. All the characteristics of the experimental *F-V* relation (Figure 5A,D) and *P-F* relation (Figure 5B,E) can be recovered by (i) increasing the forward and backward rate constants of step 1 (k_1_ and k_−1_), by ~6 times for the mammalian muscle (red in Figure 8A in Methods) and by ~3 times for the frog muscle (red in Figure 8E in Methods) and (ii) increasing the maximum value of k_3_ attained during shortening by 60% for the mammalian muscle (red in Figure 8C in Methods) and by a factor of 2 for the frog muscle (red in Figure 8G in Methods). In this case, however, while φ_0_ is in agreement with the experimental value, φ*_P_*_max_ (identified by the red arrow in Figure 5C,F) increases with respect to φ_0_ by ~6-fold for the mammalian muscle (Table 3, third line) and by ~10-fold for the frog muscle (Table 4, third line), that is at least twice the observed increases. Thus, by opportunely tuning the strain dependence of the rate constant for motor detachment, so as to preserve the relatively low value of k_3_ in the range *d* (the relative axial position between the motor and the actin monomer to which it is attached) explored in isometric conditions, Model 3 can predict the isometric energy rate (Table 1 and Table 2), but fails to predict the limited increase in energy rate that characterizes the isotonic contraction at the maximum power.

**Figure 5 ijms-24-12324-f005:**
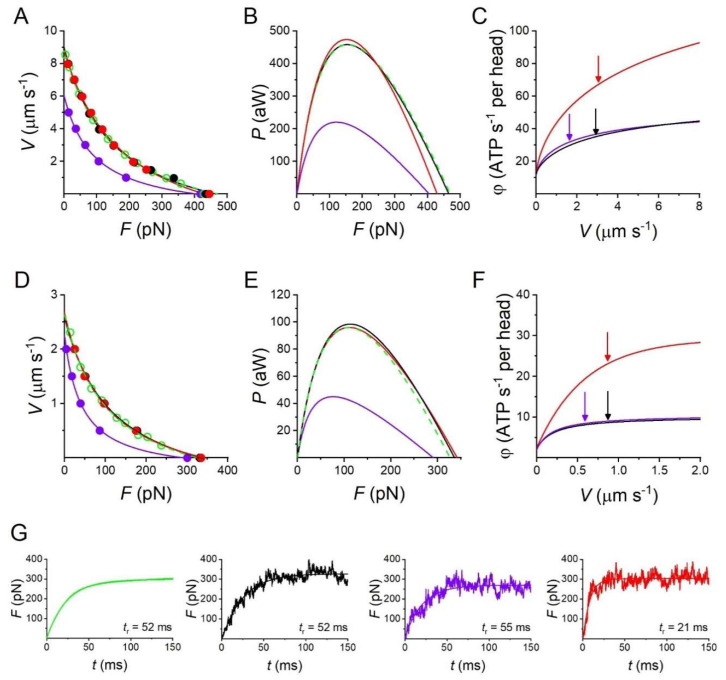
Model simulation of the performance of the half-thick filament of the mammalian skeletal muscle at 25 °C and frog muscle at 4.6 °C. The model with slipping (Model 1, black) fits the experimental data (green) of either the mammalian muscle (**A**–**C**) or the frog muscle (**D**–**F**). (**A**,**D**) *F*-*V* relation; (**B**,**E**) *P-F* relation_;_ (**C**,**F**) φ–*V* relation, where the black arrow identifies φ*_P_*_max_. In Model 2 (violet, without slipping) *P*_max_ decreases by a factor of 2 (**B**,**E**), without change in φ*_P_*_max_ ((**C,F**), violet arrow). In Model 3 (red), *P*_max_ is recovered ((**B**,**E**), red) by raising the attachment/detachment rate constants according to Figure 8 in Methods (black to red), but this implies that φ*_P_*_max_ ((**C**), red arrow) almost doubles, becoming ~6·φ_0_ and ~10·φ_0_, in the mammalian and frog muscle, respectively. (**G**) Green: force redevelopment from zero following a 5% rapid shortening superimposed on the isometric contraction of a frog muscle fibre (from Figure 3a green in [38]). Black, violet and red, simulated force redevelopment with the model identified by the colour according to the code as above.

A mechanical parameter that is sensitive to the differences in the kinetic scheme between Model 1 and Model 3 is the rate of transition of force to the steady-state isometric value *F*_0_. In fact, the time course of force rise depends on the rate constants governing attachment (step 1) and force generation (step 2), and, specifically in this model, on step 1 that is the rate-limiting step in the process. The experimental signal to be simulated cannot be the initial force development following the start of stimulation, which is influenced by the time taken for thick filament activation (Figure 3 black in [38]). An adequate signal, reliably available only from frog fibre mechanics, is the time course of force redevelopment following a 5% rapid shortening superimposed on the isometric contraction (Figure 3a green in [38]). In fact, in this case, the release size is enough to drop the force and keep it at zero, avoiding that the recovery of the thick filament OFF state progressing with the time the force is kept at zero [39] contaminates the time course of force redevelopment slowing it down.

The rise time *t*_r_ of force development (the time from 10 to 90% of *F*_0_) estimated on the exponential fit to the recorded trace (Figure 5G green) is 52 ms. *t*_r_ simulated by Model 1 (Figure 5G black, average of 12 traces) is 52 ± 9 ms, while *t*_r_ simulated by Model 3 (Figure 5G red, average of 12 traces) is 21 ± 3 ms, demonstrating that the kinetic scheme of Model 3 fails to predict the relatively low rate of transition to the steady state. Notably, *t*_r_ simulated by Model 2 (Figure 5G violet, average of 13 traces) is 55 ± 11 ms, which is not significantly different from *t*_r_ measured on the experimental trace. This is the consequence of the fact that Model 2 shares the same kinetic scheme as Model 1, apart from the slipping transition, which is effective only for the performance of the shortening muscle.

### 2.3. Mechanics and Energetics of the Sarcomere-like Nanomachine

In the previous section, the mechanical performance of fast skeletal muscle has been uniquely matched with the energetic requirements in both isometric and isotonic contractions by the assumption in Model 1 that a myosin motor can interact with two consecutive actin sites during the same ATPase cycle. This mechanism is conceivable as an emergent property of the array arrangement of myosin motors in the half-sarcomere, so that a single motor can slip to the next Z-ward actin site while the other motors power actin filament sliding. However, inferring the simplified mechanism in Model 1 from muscle mechanics implies ignoring the possible contributions to muscle performance of the structural organization of the molecular motors in the three-dimensional lattice and of the other sarcomeric (cytoskeleton and regulatory) proteins.

Here, we exploit the sarcomere-like synthetic nanomachine [18] to test whether its mechanical performance can be uniquely reproduced by Model 1. The nanomachine is based on a Dual Laser Optical Tweezers (DLOT) [40], which allows the performance of an ensemble of pure isoforms of myosin motors interacting with the actin filament to be defined without the confounding effects of other sarcomeric proteins and higher hierarchical levels of organization of the muscle. In the experiment of Figure 6 (from [25]), an ensemble of eight HMM fragments purified from fast (psoas) rabbit skeletal muscle extending from the functionalized surface of an optical fibre is brought to interact with an actin filament attached with the correct polarity to a bead trapped by the DLOT acting as a force transducer (A). In solution with physiological ATP concentration (B), the interaction induces a force development to a steady maximum value that corresponds to the force generated by the muscle in isometric contraction (*F*_0_). Once the maximum isometric force is attained in the position clamp (phase 1), the control is switched to the force clamp (phase 2) and a staircase of steps to forces progressively smaller than *F*_0_ is imposed on the system to record the corresponding steady shortening velocities (phases 3–5). Then, the control is switched back to the position clamp (phase 6) and the force recovers to the initial *F*_0_, demonstrating that the number of motors available for attachment to the actin filament remains the same throughout the whole interaction. Figure 6C shows the *F-V* relation (green squares) and its fit with Hill’s equation (green line) superimposed with the *F-V* relations simulated by the same three models as in Figure 5A, after taking into account the methodological limits of the nanomachine (namely, the random orientation of the motors and the 100 times larger compliance in series with the nanomachine due to the trap compliance, see Methods for details) and scaling the number of available motors (*N*) from 294 per half-thick filament (Figure 5A) to 16 of the nanomachine array.

The experimental relation is properly simulated with both Model 1 (black) and Model 3 (red), while *F*_0_, *V*_0_ and *P*_max_ predicted by Model 2 (violet) are too small. Accordingly, the experimental *P-F* relation (green in Figure 6D) is reproduced by Model 1 (black) and 3 (red), while the Model 2 relation (violet) is depressed and is scaled down by a factor larger than 2 with respect to the experimental value.

The large trap compliance in series with the nanomachine (~3.7 nm pN^−1^), combined with the strain-dependent kinetics of the attached motors, causes the push–pull on the motors when the addition–subtraction of the force contribution by each motor makes actin to slide away from toward the bead, as described in detail in [18]. This affects the energy consumed in the isometric contraction and the energetic differences between isometric and isotonic contractions. In fact, Model 1 predicts that, with respect to the values in the corresponding muscle (see comparison in Table 5), φ_0_ of the nanomachine is 70% larger. Moreover, because of the depression in the step size and thus in the low-load velocity of shortening, due to the random orientation of motors (Figure 8D,H in Methods), the increase in φ*_P_*_max_ with respect to φ_0_ predicted by Model 1 for the nanomachine is smaller than for the muscle. As a consequence of these methodological limits of the nanomachine in reproducing the conditions of the half-sarcomere in vivo, the comparison of the energetic outputs of the nanomachine predicted by Model 1 and 3 is not as effective as that for the muscle for selecting the model.

The simulation of the transition of force to the steady-state isometric value *F*_0_ with the three models is shown in Figure 6E. In the nanomachine, the force rise is slowed down by the large in series compliance, but this limit is taken into account in the simulation of the nanomachine performance, recovering the possibility to compare the effects of the rate constants governing attachment on the force rise. The rise time *t*_r_ of force development estimated from the exponential fit to the recorded trace (Figure 6E green) is 200 ± 18 ms (n = 16). *t*_r_ simulated by Model 1 (Figure 6E black, average of 18 traces) is 211 ± 8 ms, while *t*_r_ simulated by Model 3 (Figure 6E red, average of 15 traces) is 55 ± 3 ms, demonstrating that the kinetic scheme of Model 3 fails to predict the relatively low rate of transition to the steady state. Notably, *t*_r_ simulated by Model 2 (Figure 6E violet, average of 12 traces) is 185 ± 6 ms, which is not significantly different from *t*_r_ measured on the experimental trace. Thus, also for the simplified motor system reproduced by the nanomachine, the slipping transition for the muscle is not effective in isometric condition, and the slipping transition is an energetically well-suited mechanism that emerges only to enhance the performance of the contraction during shortening.

## 3. Discussion

In this paper, a careful comparative analysis of the mechanics and energetics of fast skeletal muscle myosin in vivo and in the synthetic half-sarcomere-like nanomachine is reported. The integrated information is used to constrain a unique three-state mechanokinetic model of the ATP-fueled actin–myosin interaction. The conclusions hold several important and original achievements toward a detailed understanding of the molecular basis of muscle performance and the emergent properties of the sarcomere structure.

Matching mechanical performance and energy consumption in fast skeletal muscle implies that the observed maximum power output during shortening is accounted for by an energy rate larger than the energy rate in the isometric contraction only by a factor of two for the mammalian muscle and of four for the frog muscle. First of all, it must be noted that the corresponding limited increase in the ATPase rate during shortening cannot be related to the rise in a fraction of myosin motors recovering the OFF state (in which ATP hydrolysis is inhibited [41,42]), induced by the reduction in force below the isometric value. In fact, it has been found that, during contraction, the OFF state is progressively recovered only during prolonged unloaded shortening [38,39]. Notably, in this latter case, the ATPase was found to progressively decrease [43].

The chemomechanical energy transduction cycle that accounts for the corresponding limited increase in the ATPase rate during shortening at the maximum power is identified with a model simulation (Model 1) that assumes that, during shortening, an actin-attached myosin motor that has partly gone through the working stroke is able to slip to the next actin monomer 5.5 nm farther from the sarcomere centre (Figure 4) and, after a strain-dependent re-equilibration of its state toward the beginning of the working stroke, promote further shortening by the completion of the working stroke, followed by ADP–ATP exchange in its catalytic site and detachment. The thermodynamic compatibility of this process within the free energy change accounted for by the hydrolysis of one molecule of ATP has been demonstrated in a previous study [35], as reported in Appendix A.

The power of the slipping hypothesis in Model 1 in matching the mechanical performance and the energetic requirements of fast skeletal muscle, and specifically, the limited increase in energy rate moving from the isometric contraction to the isotonic contraction at the maximum power, is unique. Evidence for this is given by testing the alternative model (Model 3), in which the slipping process is suppressed and the rate constants of the relevant steps of the conventional actin–myosin interaction cycle (attachment and detachment) are adapted to simulate the experimental *F-V* and *P-F* relations. Model 3 implies that in moving from the isometric contraction to the isotonic contraction at *P*_max_, the energy rate increases by 6 and 10 times in mammalian and frog muscle, respectively, twice the observed increases.

A further striking limit intrinsic to Model 3 is that the increase in the rate constant for the attachment step required to fit the maximum power implies the increase in the rate of force development that becomes three to four times larger than that observed. In this respect, it must be noted that the rate of force development can be kept as low as required, also assuming a specific geometrical hindrance in the isometric condition consequent to mismatch between actin and myosin periodicities [44]. However, that model shares the contradiction of Model 3 that, to fit the power produced during shortening, the ATP hydrolysis rate increases by eight to nine times with respect to the isometric rate.

We conclude that the mechanical performance of shortening muscle, within the limits imposed by the energetics, can be predicted only by assuming that a myosin motor uses two consecutive actin monomers to complete its ATPase cycle, as described in Model 1. This is an emergent property of muscle myosin working in ensemble that is evident even in a simplified actin–myosin system as the synthetic nanomachine.

In contrast with the above conclusion, it has been recently shown that the *F-V* relation of the nanomachine can be fit with a conventional model implying 1:1 coupling between mechanical and biochemical steps [45]. That model simulation, however, does not take into account the limits of the nanomachine in reproducing the mechanical performance that the motor array would have generated if the motors were correctly oriented and had a 100 times smaller (sarcomere-like) compliance in the series. A correct definition for the ability of a kinetic scheme to fit the nanomachine output should introduce in the simulation, as in ours, the methodological limits of the nanomachine. Conversely, that model would not be able to predict the mechanical performance of both the nanomachine with the correctly oriented motors and the muscle of origin.

The finding that a unique kinetic model (Model 1) fits both mechanics/energetics of skeletal muscle and the nanomachine mechanics is a striking demonstration of the power of the nanomachine to reproduce the mechanical performance and the underlying biochemistry and energetics of the skeletal muscle from which the myosin motors are extracted. A more quantitative test is carried out in Figure 7, showing how the simulated relevant parameters of the mechanical performance of skeletal muscle (*F*_0_, *P*_max_ and *V*_0_, dots) scale down from the value characteristics of the native half-thick filament to those measured with the synthetic nanomachine (dashed lines) using the number of motors available for the actin interaction (*N*) as the only free parameter. *N* is scaled down from that in the half-thick filament, 294 [37], but for simplicity, the relations are shown only for *N* ranging from 42 to 10. For *N* = 16, all the three relations intersect the experimental value (dashed lines, 16.79 ± 0.38 pN for *F*_0_, 5.45 aW for *P*_max_ and 3.54 ± 0.13 µm s^−1^ for *V*_0_); 16 is the value of *N* expected from the number of rupture events in rigor in these experiments (8, [18]), considering that each motor of the myosin dimers identified in rigor behaves independently in 2 mM ATP.

The mechanical manifestation of the working stroke in the attached motors in situ is the rapid force recovery following a stepwise shortening superimposed on the isometrically contracting muscle fibre [36]. With this method, it was estimated that the maximum sliding distance accounted for by the working stroke is 12 nm [34,46], in agreement with the size of the working stroke suggested by the crystallographic model [4]. Double steps superimposed on the isometric contraction of single muscle fibres showed that the working stroke is regenerated within a time much faster than that accounted for by φ_0_ [33], a finding confirmed with X-ray diffraction showing a similar rapid rate of regeneration for the conformation of the myosin motors characteristic of the early state of the working stroke [47,48]. Accordingly, also in Ca^2+^-activated skinned fibres of rabbit psoas, it was found that the velocity transient elicited by a stepwise change in force could be explained by the generation of a second working stroke faster than expected from the ATPase rate [35,49]. Single molecule manipulation with a microneedle provided independent and direct evidence for the generation of multiple working strokes by an actin-attached myosin motor during an ATP hydrolysis cycle [50,51]. All those experiments offer further kinetic and structural constraints for the development of more detailed models able to explain, as in Model 1, the matching between mechanical performance and energy consumption by myosin.

## 4. Material and Methods

### 4.1. Experimental Data

Mechanical and energetic data reported are acquired according to methods detailed in the referred papers.

### 4.2. Model Simulation

The mechanical and energetic characteristics collected from experiments in situ (muscle) and in vitro (nanomachine) are numerically simulated with a stochastic model already published [18]. The model estimates the probability distributions of potential results by allowing for random variation in inputs over time until the Standard Deviation of the result is lower than 5% of the mean value obtained from single cycles of 5 s iterations. The mechanical cycle of the motors is depicted in Figure 4. The state transitions as well as the strain of the attached motors are stochastically determined according to the kinetics reported in Figure 8. Each attached motor exerts, on the actin filament, a force that depends on its conformation and its position with respect to the actin monomer to which it is attached. Details on the calculation are given in [18].

**Figure 8 ijms-24-12324-f008:**
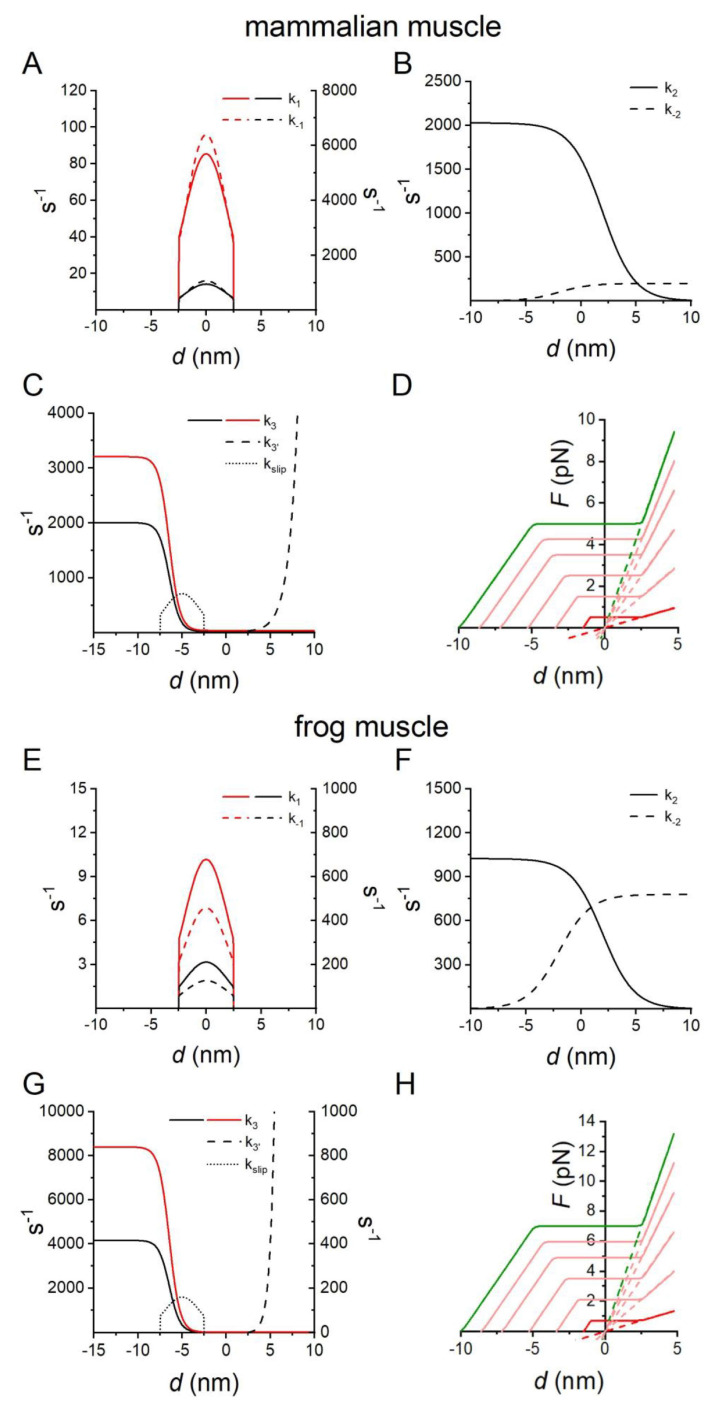
Kinetic and mechanical features of the actin–myosin cycle. (**A**–**D**) mammalian muscle, (**E**–**H**) frog muscle. (**A**–**C**,**E**–**G**) Dependence of the rate functions for the state transitions on *d* (the relative axial position between the motor and the actin monomer, with *d* = 0 when the force in A1 is zero). Where the transition is reversible, k_i_ is the forward and k_−i_ is the backward rate constant. Black lines, Model 1, according to the model published in [18,25]. Red lines, Model 3, with the relevant rate functions enhanced in order to fit the force–velocity relation and *P*_max_ without slipping. (**A**,**E**) step 1, attachment reaction: k_1_ continuous line, left ordinate; k_−1_ dashed line, right ordinate. (**B**,**F**) Step 2, force-generating transition: k_2_ continuous line, k_−2_ dashed line. (**C**,**G**) Step 3, detachment reaction, k_3_ continuous line. The slipping of A2 to the next actin farther from the centre of the sarcomere is governed by k_slip_ (dotted line). Detachment from A2 following slipping, k_3′_ dashed line. (**D**,**H**) Force profile of the attached states of the motor as a function of *d*. A1 state, dashed line and A2 state, continuous line. Green line, force profile of correctly oriented motors; red line, force profile of motors 180° away from the correct orientation; pink lines, four profiles corresponding to four intermediate motor orientations. Panel (**D**) from [25].

The kinetic scheme is based on three states of the myosin motors (one detached, D, and two force-generating attached states, A1 and A2 (Figure 4)). The stiffness of the correctly oriented attached motors is 2 pN nm^−1^ [18], obtained as an average from [19,52], and the kinetics of the state transitions (Figure 8A–C for mammalian muscle, Figure 8E–G for frog muscle) are selected on the basis of the mechanical and energetic properties of the corresponding intact muscles, as detailed in the text.

Detached motors (D), with the hydrolysis products (ADP and inorganic phosphate, P_i_) in the catalytic site, attach to the actin monomer (5 nm in diameter) (D → A1, step 1) at values of *d* (the relative axial position between the motor and the actin monomer with *d* = 0 when the force in A1 is zero) ranging from −2.5 to 2.5 nm, according to the principle of the nearest-neighbour interaction. Following the state transition A1 → A2 (step 2), the motors undergo the working stroke that accounts for the generation of force at *d* = 0 and its maintenance during shortening [34,36]. The working stroke, accompanied by Pi release, is a multistep reaction, the extent and speed of which depend on the degree of shortening and external load [34,35]. During isotonic shortening, the speed of the working stroke is fast enough for the reaction to be considered at the equilibrium and the A2 force to follow the *d*-dependence shown in Figure 8D (mammalian muscle) and Figure 8H (frog muscle), green continuous line [34,36]. In the nanomachine, the depressant effect of the random orientation of motors on the force and the step size [51,53] is taken into account by reducing the force and sliding distance at which the A2 motor maintains the force from the values exhibited by the correctly oriented motor (*F*_c_ = 5 pN and *L*_c_ = 10 nm, respectively) in proportion to the deviation from correct orientation (pink continuous lines) up to a minimum of 0.5 pN and 1.5 nm, respectively (red continuous line), for a deviation of 180°. The instantaneous stiffness of A1 motors reduces accordingly (dashed lines) and becomes 0.2 pN nm^−1^ for the orientation at 180° deviation. The A2 motors detach (A2 → D, Figure 4, step 3) following ADP release and ATP binding with a speed that, under physiological (ATP) concentration (≥2 mM), is dictated by the conformation-dependent kinetics of ADP release [35,54,55]. In order to maintain the scheme simple, the hydrolysis step and the repriming of the working stroke are incorporated in state D and contribute to the limited speed of the attachment reaction.

## 5. Conclusions

An emergent property of muscle myosin working in ensemble, demonstrated in either the muscle sarcomere or the unidimensional nanomachine, is that the mechanical performance of shortening muscle can be accounted for by energetics only by assuming that a myosin motor can use two consecutive actin monomers to complete its ATP-ase cycle.

## Figures and Tables

**Figure 1 ijms-24-12324-f001:**
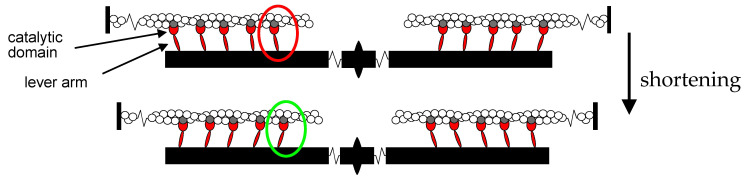
Cartoon representing sarcomere shortening induced by the working stroke in the myosin motor. Reciprocal sliding between the actin filaments (white), originating from the Z line, and the myosin filaments (black), originating from the M line and carrying the two antiparallel arrays of myosin motors (red, all in the same configuration for the sake of simplicity), is powered by the working stroke in the actin-attached myosin motors (red to green circle). The working stroke consists in the interdomain tilting of the lever arm about the catalytic domain firmly attached to the actin site (grey).

**Figure 2 ijms-24-12324-f002:**
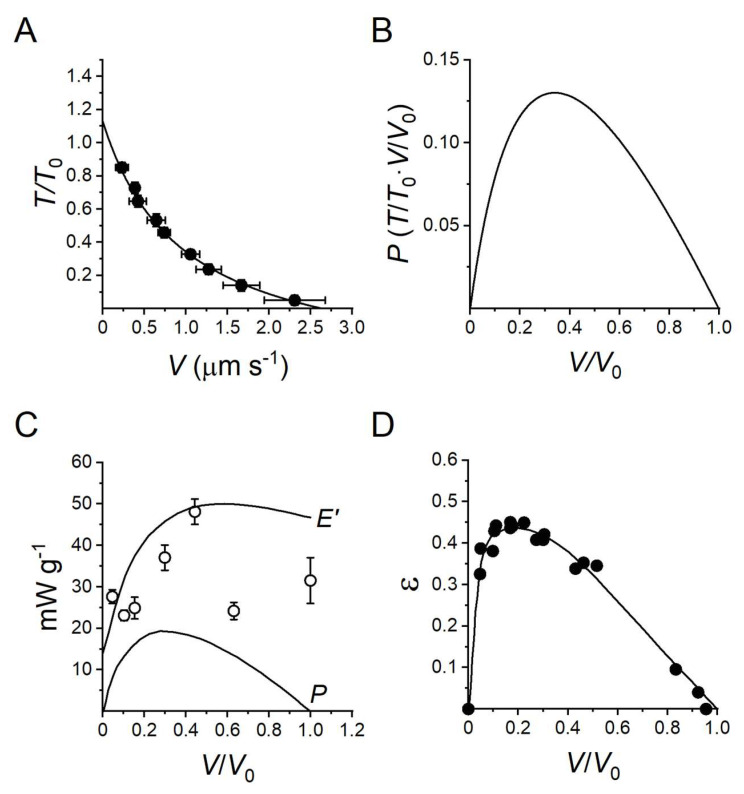
Dependence on shortening velocity of force, rate of energy liberation and efficiency of frog muscle. (**A**) Force–velocity (*T-V*) relation from single muscle fibres of *Rana esculenta* at 4.6 ± 0.9 °C (mean ± SD) and 2.15 μm SL. *T* is relative to the isometric plateau value, *V* is per half-sarcomere. The points are the means ± SEM of data obtained from 15 fibres grouped in classes of *V* 0.5 μm s^−1^ wide. Bars indicating the SEM are not visible when their size is smaller than symbol size. The average isometric force *T*_0_ is 164 ± 32 kPa (mean ± SD). The continuous line is Hill’s hyperbolic equation fitted to data (black circles). Hill’s parameters (±SEM) are *a*/*T*_0_ = 0.36 ± 0.10, *V*_0_ = 2.65 ± 0.14 μm s^−1^, *T*_0_*/*T*_0_ (the intercept on the ordinate) = 1.13 ± 0.09. (**B**) Power–velocity (*P*-*V*) relation in relative units calculated from Hill’s fit in (**A**). (**C**) Dependence on *V* of the rate of energy liberation (*E*′ = *P* + *Q*′, upper line) and of *P* (lower line) in *sartorius* muscle of *Rana pipiens* at 0 °C. White circles are the rate of energy calculated from the rate of ATP splitting (φ, mean ± SEM). From [15] as reported by [10]. (**D**) Mechanical efficiency (ε) as defined in the text (black circles, data). Adapted from [16]. Note that the efficiency is slightly lower than that obtained from data reported in Table 1 from a more recent review [17].

**Figure 3 ijms-24-12324-f003:**
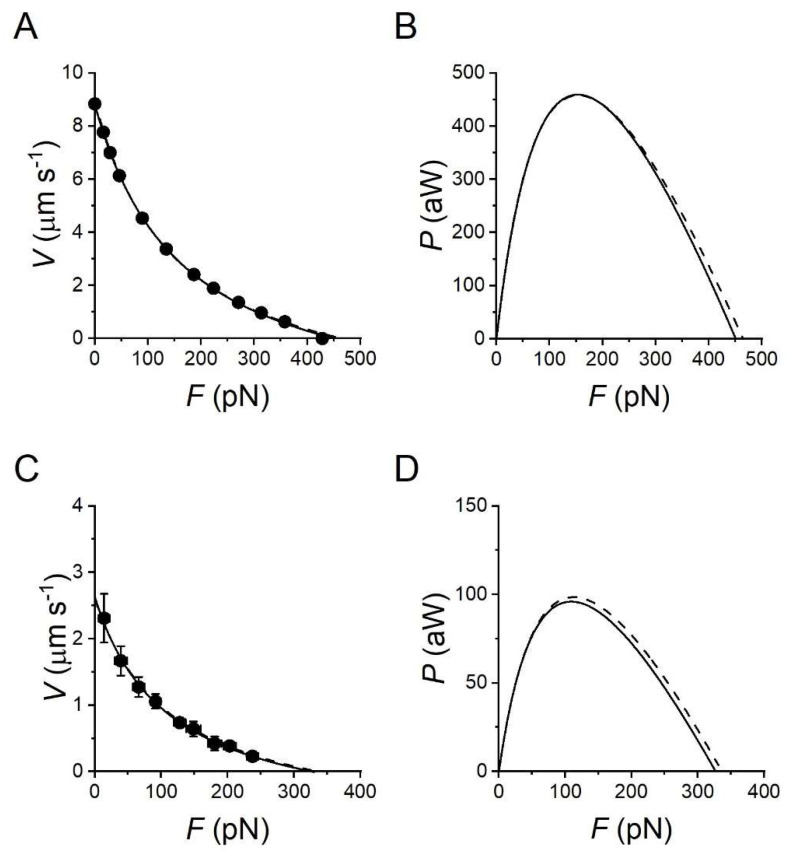
Mechanical performance per half-thick filament of the EDL muscle of the rat at 25 °C in comparison to that of the frog muscle fibre at 4.6 °C. (**A**) Force–velocity relation of the EDL muscle. The continuous line is Hill’s hyperbolic equation fitted to the experimental force–velocity relation of [21]. Force is scaled at the level of the half-thick filament (*F*), with *T*_0_ = 245 kPa [20]. *V* in μm s^−1^ per half-sarcomere is calculated from *V*_0_ = 7.3 muscle length per s [21] with SL = 2.4 μm. Hill’s parameter *a*/*F*_0_ is 0.37: the dashed line is Hill’s hyperbolic equation fitted to the results of the simulation with Model 1 as detailed in Figure 4. (**B**) *P*-*F* relation calculated from Hill’s fit on the experimental data in (**A**) (continuous line) and from the simulation (dashed line). A and B from Figure 3b,c in [25]. (**C**,**D**) Continuous lines: force–velocity relation and power–force relation per half-thick filament of the frog muscle fibre, from data in Figure 2A and Figure 2B respectively. Dashed lines: results of the simulation with Model 1.

**Figure 4 ijms-24-12324-f004:**
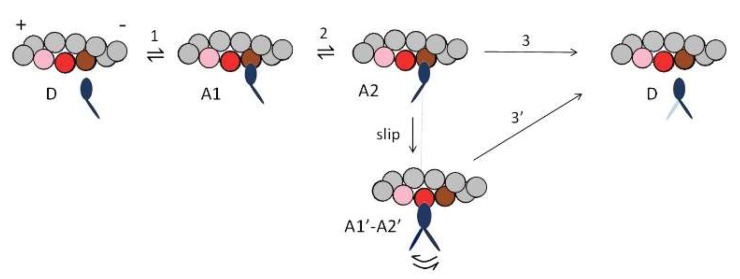
Model 1. Kinetic scheme from Figure 4A of [18], with three states of the myosin motor (blue): D, detached; A1 and A2, attached to an actin monomer (brown). During shortening, the motor attached in the A2 state can slip to the next actin monomer farther from the centre of the sarcomere (red) within the same ATPase cycle. The probability of a second slipping to the pink monomer is limited to 1/10 of the first slipping. Gray, actin monomers not interacting with the myosin motor.

**Figure 6 ijms-24-12324-f006:**
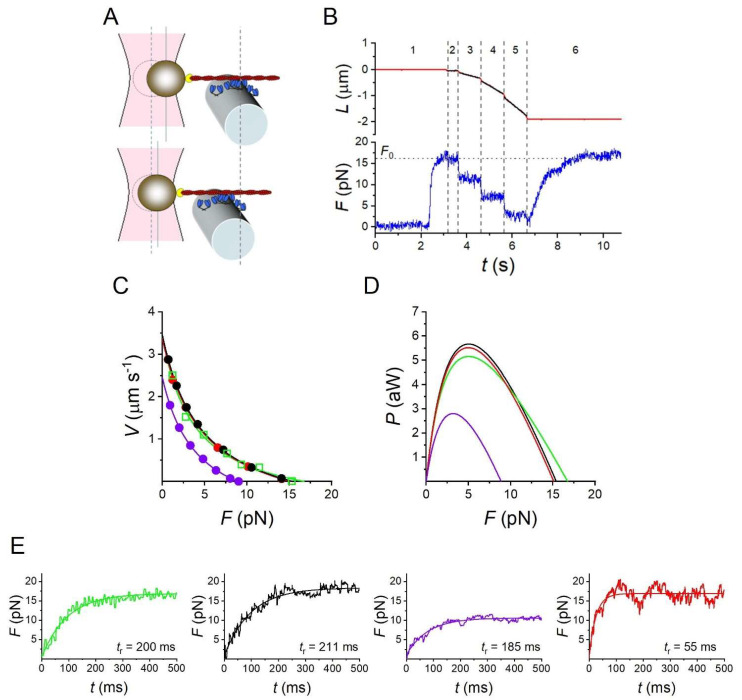
Model simulation of the outputs of the nanomachine powered by 16 myosin motors from rabbit psoas at 25 °C. (**A**) Schematic representation of two snapshots during the interaction between the actin filament and the motor ensemble. Upper panel: in position clamp at *F*_0_; lower panel: in force clamp at 0.4 *F*_0_. (**B**) Recording of the actin filament sliding (*L*, upper trace, red) and force (*F*, lower trace, blue) during an interaction. The numbers indicate the phases as described in the text. (**C**) *F*-*V* relations from the experiment (green square) and its interpolation by Hill’s equation (green line) and from the simulations with Model 1 (black), Model 2 (violet) and Model 3 (red) as detailed in the text. (**D**) Experimental and simulated *P-F* relations with the same colour code as in (**C**). (**E**) Force development in position clamp (noisy trace) and its fitting with an exponential, either recorded (green), or simulated with Model 1 (black), Model 2 (violet) and Model 3 (red). The respective rise times (*t*_r_), calculated as defined in the text, are reported next to the trace. Panels (**A**,**B**) are from [25].

**Figure 7 ijms-24-12324-f007:**
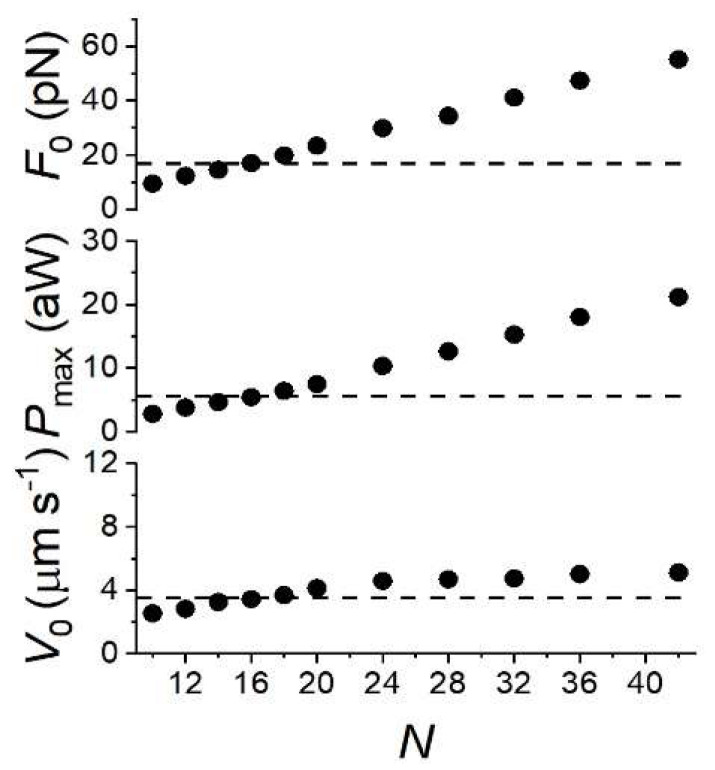
Dependence on *N* of the simulated *F*_0_, *P*_max_ and *V*_0_ (dots), the three parameters featuring the nanomachine performance, as indicated in the ordinate of each plot. The dashed lines indicate the respective experimental values recorded from the nanomachine in 2 mM ATP. The nanomachine was powered by an array of rabbit psoas HMM molecules deposited on the functionalized surface that gave a number of rupture events = 8.2 ± 1.2 in the ATP-free solution. Adapted from Figure 3 in [25].

**Table 1 ijms-24-12324-t001:** Energetic properties of sartorius muscle of *Rana temporaria* at 0 °C. Contributions to energy rate from sources different from myosin motors are subtracted. From [17] and references therein.

	*T* _0_	1/3 *T*_0_
*E*′ (mW g^−1^)	11.2	45.0
*P* (mW g^−1^)	0	22.0 (*P*_max_)
ATPase rate (μmol s^−1^ g^−1^)	0.23	0.90
φ (s^−1^)	1.1	4.3

**Table 2 ijms-24-12324-t002:** Energetic properties of fast mammalian muscle. The energy rates of the isometric contraction (*E*′_0_) and during shortening at *V*_Pmax_ (*E*′_Pmax_) measured in the muscle (first two columns) are translated to the corresponding φ’s for comparison with skinned fibre data using a Δ*G*_ATP_ of 60 kJ mol^−1^ [17] and a motor concentration of 0.18 mM [27]. The paper originating the data is indicated by the superscript number. In the case of [30], *P*_max_, in brackets, is calculated from η.

		*E*′_0_ (mW g^−1^)	*E*′_Pmax_ (mW g^−1^)	*P*_max_ (mW g^−1^)	ε, η	φ_0_ (s^−1^)	φ*_P_*_max_ (s^−1^)	φ*_P_*_max_/φ_0_
Mouse EDL, intact [26]	21 °C	134	214	57.2	0.28	12.4	19.8	1.6
Mouse EDL, intact [27]	25 °C	144	269	70	0.26	13.3	25	1.87
Rat IIB, skinned [28]	12 °C	15	40.2	9.59	0.24	1.39	3.7	2.7
Rabbit psoas, skinned [29]	15 °C					2.1	6	2.8
Rabbit psoas, skinned [30]	10 °C	11.4	30	(10)	0.33	1.05	2.77	2.6
Human IIA, skinned [31]	20 °C	48	96	30	0.31	4.4	8.89	2

**Table 3 ijms-24-12324-t003:** Model simulation of mechanical and energetic parameters of the mammalian muscle at 25 °C. Model 1, upper line, implies the possibility of the attached motor in state A2 to slip to the next Z-ward actin site, as detailed by the set of rate constants in black Figure 8A,C in Methods. Model 2, middle line, is the same kinetic scheme as Model 1 without the slipping possibility. Model 3 attachment and detachment rate constants are increased (red in Figure 8A,C in Methods), so as to fit the observed curvature of the force *F-V* relation and the maximum power as in Figure 3A,B. The parameters for the half-thick filament are in agreement with the corresponding values of the reference in brackets: *N* (49 crowns × 6 motors per crown =) 294 [37]; *F*_0_, isometric force [20]; *r*_0_, isometric duty ratio [21]; φ_0_, flux through step 1 of the cycle in isometric condition, corresponding to the ATP hydrolysis rate per myosin head at *F*_0_ (Table 2); *a*/*F*_0_, the Hill’s parameter expressing the curvature of the *F-V* relation [20]; *V*_0_, maximum shortening velocity [21]; *P*_max_, maximum power; φ*_P_*_max_, ATP hydrolysis rate per myosin head at *P*_max_ (Table 2).

*N* = 294	*F*_0_ (pN)	*r* _0_	φ_0_ (s^−1^)	*a*/*F*_0_	*V*_0_ (µm s^−1^)	*P*_max_ (aW)	φ*_P_*_max_ (s^−1^)
Model 1	433 ± 5	0.32	11.65	0.36	8.61 ± 0.16	462	35.5
Model 2	418 ± 6	0.31	11.71	0.22	6.02 ± 0.12	220	31.9
Model 3	445 ± 9	0.33	11.41	0.39	8.79 ± 0.14	474	67.2

**Table 4 ijms-24-12324-t004:** Model simulation of mechanical and energetic parameters of the frog muscle at ~5 °C. Model 1, upper line, implies the possibility of the attached motor in state A2 to slip to the next Z-ward actin site, as detailed by the set of rate constants in black in Figure 8E,G in Methods. Model 2, middle line, is the same kinetic scheme as Model 1 without the slipping possibility. Model 3 attachment and detachment rate constants are increased (red in Figure 8E,G in Methods), so as to fit *F-V* relation and *P-F* relation as in Figure 3C,D. The parameters for the half-thick filament (294 motors), *F*_0_, *a*/*F*_0_, *V*_0_, and *P*_max_, are in agreement with the corresponding experimental values in Figure 3C,D; *r*_0_, isometric duty ratio [6]; φ_0_, flux through step 1 of the cycle in isometric condition, corresponding to the ATP hydrolysis rate per myosin head at *F*_0_; φ*_P_*_max_, ATP hydrolysis rate per myosin head at *P*_max_ (Table 1).

*N* = 294	*F*_0_ (pN)	*r* _0_	φ_0_ (s^−1^)	*a*/*F*_0_	*V*_0_ (µm s^−1^)	*P*_max_ (aW)	φ*_P_*_max_ (s^−1^)
Model 1	336 ± 12	0.24	2.30	0.34	2.60 ± 0.07	98.5	8.63
Model 2	291 ± 23	0.23	2.10	0.14	2.25 ± 0.03	45.0	8.45
Model 3	342 ± 21	0.24	2.40	0.30	2.67 ± 0.11	96.0	23.1

**Table 5 ijms-24-12324-t005:** Model 1 simulation of mechanical and energetic parameters of the mammalian muscle half-thick filament and of the nanomachine at 25 °C. Adapted from [25].

	*F*_0_ (pN)	*r* _0_	φ_0_ (s^−1^)	*a*/*F*_0_	*V*_0_ (µm s^−1^)	*P*_max_ (aW)	φ*_P_*_max_ (s^−1^)
*N* = 294 Compliance 0.01 nm pN^−1^	433 ± 5	0.32	11.65	0.36	8.61 ± 0.16	462	35.50
*N* = 16 Compliance 3.7 nm pN^−1^ + random	15.8 ± 0.4	0.40	18.21	0.24	3.45 ± 0.13	5.45	26.21

## Data Availability

All relevant data, associated protocols, and materials are within this paper or in the literature quoted as indicated in References. If any additional information is needed, it will be available upon request from the corresponding author.

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
