# Peer review of "Matching Mechanics and Energetics of Muscle Contraction Suggests Unconventional Chemomechanical Coupling during the Actin–Myosin Interaction"

_ijms, 2023, doi:10.3390/ijms241512324_

Round 1

Reviewer 1 Report

This manuscript reviews the literature describing the coupling between the biochemistry and the mechanics of muscle contractions. It proposes several models to reconcile these events and then provides a test of the system using a in vitro nanomachine that consists of an optically trapped actin filament interacting with a limited numbers of myosin fragement.

My only major concern is that the authors have not considered the possibility that a significant fraction of the myosin heads in the muscle may be held in a. conformation (typically referred to as the interacting head motif) wherein such a myosin is not capable of interacting with actin and would not be contributing significantly to the amount of ATP hydrolyzed.  How would this consideration affect their model?

Just a few minor points. 

The term HMM is used in a few places, but I don't see where it was defined.

In Fig 6 legend, panel E is referred to as panel D

In general the manuscript reads well, but there are a few instances of wrong tenses being used, etc.

Author Response

Reply to the Editor

We are submitting the revised version of the manuscript ID ijms-2502819. All the recommendations of the Editor and the reviewers reported here (in italic) have been considered as detailed in the following point by point replies. The revisions on the text are highlighted as requested.

(I) Please check that all references are relevant to the contents of the
manuscript
.

Done

(II)  Any revisions to the manuscript should be highlighted, such that any
changes can be easily reviewed by editors and reviewers.

Done

(III) Please provide a cover letter to explain, point by point, the details
of the revisions to the manuscript and your responses to the referees’
comments.

Done

(IV) If you found it impossible to address certain comments in the review
reports, please include an explanation in your appeal.

Our concerns about some  comments of reviewer #2 are reported in the reply

If one of the referees has suggested that your manuscript should undergo
extensive English revisions, please address this issue during revision.

Reviewer #1 pointed at “a few instances as wrong tenses”. We had done a careful revision of the language and amended all possible errors in the use of tenses.

Reply to  Reviewers

Reviewer #1

This manuscript reviews the literature describing the coupling between the biochemistry and the mechanics of muscle contractions. It proposes several models to reconcile these events and then provides a test of the system using a in vitro nanomachine that consists of an optically trapped actin filament interacting with a limited numbers of myosin fragement.

My only major concern is that the authors have not considered the possibility that a significant fraction of the myosin heads in the muscle may be held in a. conformation (typically referred to as the interacting head motif) wherein such a myosin is not capable of interacting with actin and would not be contributing significantly to the amount of ATP hydrolyzed.  How would this consideration affect their model?

We thank the reviewer for promoting the clarification relative to the possible influence of the degree of thick filament activation and of the recently discovered mechanosensing mechanism suggesting that the ATPase in active muscle can be modulated by different mechanical conditions that imply different stress on the thick filament. We have now clarified in the revised version of the paper (page 16) that a thick filament activation dependent modulation of the ATPase is excluded in the isometric and isotonic contractions considered here and  occurs only during prolonged unloaded shortening for the progressive recovery of the OFF state (Linari et al., 2015; Fusi et al., 2017; Homsher et al 1981). 

Just a few minor points. 

The term HMM is used in a few places, but I don't see where it was defined.

Thanks to the reviewer the term HMM is now defined at the first place it appears in the text

In Fig 6 legend, panel E is referred to as panel D

Corrected

Comments on the Quality of English Language

In general the manuscript reads well, but there are a few instances of wrong tenses being used, etc.

Tenses corrected and other minor edits done, wherever necessary, following a careful analysis

Reviewer 2 Report

Thanks for your manuscript to submit the Jounal. However, this need to be revised by author many points again. 

1) I could not understand this type of the manuscript.

2) What is this manuscript's purpose? Could you explain more exactly for readers.  

3) I think that this manuscript was not write based on the ins traction for authors in this Journal. 

4) In addition, selection of the references and subject are not clear, this will need to revise more and more.

5) So, we will not be able to evaluate this manuscript in the present form. 

Author Response

Reply to the Editor

We are submitting the revised version of the manuscript ID ijms-2502819. All the recommendations of the Editor and the reviewers reported here (in italic) have been considered as detailed in the following point by point replies. The revisions on the text are highlighted as requested.

(I) Please check that all references are relevant to the contents of the
manuscript
.

Done

(II)  Any revisions to the manuscript should be highlighted, such that any
changes can be easily reviewed by editors and reviewers.

Done

(III) Please provide a cover letter to explain, point by point, the details
of the revisions to the manuscript and your responses to the referees’
comments.

Done

(IV) If you found it impossible to address certain comments in the review
reports, please include an explanation in your appeal.

Our concerns about some  comments of reviewer #2 are reported in the reply

If one of the referees has suggested that your manuscript should undergo
extensive English revisions, please address this issue during revision.

Reviewer #1 pointed at “a few instances as wrong tenses”. We had done a careful revision of the language and amended all possible errors in the use of tenses.

Reply to  Reviewers

Reviewer # 2

Thanks for your manuscript to submit the Jounal. However, this need to be revised by author many points again. 

It has been hard to account for the comments of the reviewer also because of the corrupted English language used, however we have tried our best.

1) I could not understand this type of the manuscript.

In the web page of the journal under the point “scope” it is reported: “Fundamental theoretical problems of broad interest in biology, chemistry and medicine”. By the way the argument of this paper was preliminarily proposed to the Editor and got his/her agreement.

2) What is this manuscript's purpose? Could you explain more exactly for readers.

The answer is in the first sentence of the abstract: “The mechanical performances of the vertebrate skeletal muscle during isometric and isotonic contractions are interfaced with the corresponding energy consumptions to define the coupling between mechanical and biochemical steps in the myosin-actin energy transduction cycle”.  

3) I think that this manuscript was not write based on the ins traction for authors in this Journal.

If we have correctly extracted the question from the corrupted sentence, the answer is the same as at point 1

4) In addition, selection of the references and subject are not clear, this will need to revise more and more.

We do not understand what the referee means, unless ignorance of the subject and of the related literature

5) So, we will not be able to evaluate this manuscript in the present form. 

The problems of the reviewer (now shared with somebody else, unless he/she moves to majestic plural) in relation to the manuscript could be explained as suggested in the reply to the point above.  

Round 2

Reviewer 2 Report

I think that this manuscript is not will be able to accept.